# Decídetexto: Feasibility and Acceptability of a Mobile Smoking Cessation Intervention in Puerto Rico

**DOI:** 10.3390/ijerph18041379

**Published:** 2021-02-03

**Authors:** Francisco Cartujano-Barrera, Cristina I. Peña-Vargas, Evelyn Arana-Chicas, José G. Pérez-Ramos, Josiemer Mattei, Alejandra Hurtado-de-Mendoza, Rosario Costas-Muñiz, Julio Jiménez, Ana Paula Cupertino, Eida M. Castro

**Affiliations:** 1James P. Wilmot Cancer Institute, University of Rochester Medical Center, Rochester, NY 14642, USA; Evelyn_arana@urmc.rochester.edu (E.A.-C.); Jose_Perez-ramos@URMC.Rochester.edu (J.G.P.-R.); Paula_cupertino@urmc.rochester.edu (A.P.C.); 2Department of Psychiatry and Human Behavior, Ponce Health Sciences University, Ponce, PR 00716, USA; cpena@psm.edu (C.I.P.-V.); jcjimenez@psm.edu (J.J.); ecastro@psm.edu (E.M.C.); 3Department of Nutrition, Harvard T.H. Chan School of Public Health, Boston, MA 02115, USA; jmattei@hsph.harvard.edu; 4Lombardi Comprehensive Cancer Center, Georgetown University Medical Center, Washington, DC 20057, USA; ahd28@georgetown.edu; 5Department of Psychiatry & Behavioral Sciences, Memorial Sloan-Kettering Cancer Center, New York, NY 10065, USA; costasmr@mskcc.org

**Keywords:** Latinos, Puerto Ricans, Puerto Rico, smoking, smoking cessation, mHealth

## Abstract

The purpose of this pilot study was to assess the feasibility and acceptability of a mobile smoking cessation intervention in Puerto Rico. This was a single-arm pilot study with 26 smokers in Puerto Rico who were enrolled in Decídetexto, a mobile smoking cessation intervention. Decídetexto incorporates three integrated components: (1) a tablet-based software that collects smoking-related information to develop an individualized quit plan, (2) a 24-week text messaging counseling program with interactive capabilities, and (3) pharmacotherapy support. Outcome measures included self-reported 7-day point prevalence abstinence at Months 3 and 6, pharmacotherapy adherence, satisfaction with the intervention, and changes in self-efficacy. The average age of the participants was 46.8 years (SD 12.7), half of them (53.8%) were female. Most participants (92.3%) smoked daily and half of them (53.8%) used menthol cigarettes. All participants requested nicotine patches at baseline. However, only 13.0% of participants used the patch >75% of days. At Month 3, 10 participants (38.4%) self-reported 7-day point prevalence abstinence (88.5% follow-up rate). At Month 6, 16 participants (61.5%) self-reported 7-day point prevalence abstinence (76.9% follow-up rate). Most participants (90%, 18/20) reported being satisfied/extremely satisfied with the intervention at Month 6. Self-efficacy mean scores significantly increased from 40.4 (SD 12.1) at baseline to 57.9 (SD 11.3) at Month 3 (*p* < 0.01). The study suggests that Decídetexto holds promise for further testing among Puerto Rican smokers.

## 1. Introduction

Puerto Ricans constitute a Caribbean Latino group that, despite their U.S. citizenship status, experience clear sociodemographic disadvantages and residential segregation [1,2,3]. Moreover, Puerto Ricans suffer from marked health disparities compared to other Latino ethnic groups, despite the fact that they have higher access to government financial assistance, health insurance, and healthcare [4]. For example, Puerto Ricans in the U.S. mainland have the highest smoking prevalence and a harder time quitting smoking compared to other Latino ethnic groups [5,6,7]. Furthermore, while cigarette use varies significantly by gender among other Latino ethnic groups, cigarette use does not vary by gender among Puerto Ricans [5,8]. Additionally, smoking prevalence among Puerto Ricans varies geographically, with rates between 19–35% for Puerto Ricans in the U.S. mainland compared to 10–17% in Puerto Rico [5,8,9,10]. Despite these differences, to date only one smoking cessation study has been conducted in Puerto Rico [11]. This study concluded that combining alcohol and smoking cessation treatment did not have main effects on drinking alcohol or smoking cessation. The striking health disparities and clear gaps in the literature strongly support the need for studies to understand and address the unique tobacco-related disparities among Puerto Rican smokers in the U.S. mainland and Puerto Rico.

Evidence supports the effectiveness of mobile smoking cessation interventions [12]. Developments in the sophistication of mobile technologies allow for flexible delivery of text messages, with algorithms used to tailor content to individual motivational and behavioral needs for smoking cessation [13]. Furthermore, the impact of mobile smoking cessation interventions may be even greater among hard-to-reach and socioeconomically disadvantaged populations [12], such as Puerto Ricans. However, implementation of mobile interventions in Puerto Rico remains minimal [14,15]. The purpose of this pilot study was to assess the feasibility and acceptability of a mobile smoking cessation intervention in Puerto Rico.

## 2. Materials and Methods

### 2.1. Study Design

This was a single-arm pilot study with 26 Puerto Rican smokers who received a culturally- and linguistically-accommodated mobile smoking cessation intervention and nicotine replacement therapy (e.g., nicotine patches and gum; NRT). Participants completed an in-person baseline assessment and two telephone follow-up assessments at Months 3 and 6. Participants were compensated a $30 gift card for completing the baseline assessment, a $20 gift card for completing the Month 3 assessment, and a $50 gift card for completing the Month 6 assessment. The study was conducted between September 2019 and August 2020 at the Ponce Health Sciences University (PHSU), Puerto Rico. Study procedures were approved and monitored by the PHSU’s Institutional Review Board (protocol number 1906014547).

### 2.2. Recruitment

Recruitment started in September 2019 and ended in February 2020. Participants were recruited through clinic- and community-based efforts, including flyers, in-clinic recruitment, and word-of-mouth referrals from study participants.

### 2.3. Eligibility

Eligibility assessment was conducted over the phone. Individuals were eligible if they (1) self-identified as Hispanic or Latino, (2) knew how to read and speak English and/or Spanish, (3) were ≥21 years of age, (4) had smoked cigarettes for at least six months, (5) smoked cigarettes three or more days within a typical week, (6) reported interest in quitting smoking in the next 30 days, (7) had a cellphone with unlimited text messaging capability, (8) had a cellphone with a U.S. carrier (e.g., Verizon Wireless, AT&T, T-Mobile, Sprint, etc.), (9) knew how to send and read text messages, and (10) were willing to complete two follow-up assessments (at Months 3 and 6). Exclusion criteria included: (1) use of other tobacco products more than one day within a typical week, (2) current participation in any other smoking cessation program or use of any type of medication to quit smoking, (3) having a household member currently enrolled in the study, (4) being pregnant, breastfeeding, or planning to become pregnant in the next year, and (5) planning to move out of their current residential address in the next six months.

### 2.4. Screening and Consent

Trained research staff assessed participant eligibility. Individuals who were eligible to participate in the study were scheduled an in-person appointment by research staff. During the in-person appointment, staff discussed all aspects of study participation and confidentiality, answered any questions, and guided eligible smokers through the process of written informed consent.

### 2.5. Intervention

Decídetexto is a smoking cessation mobile intervention that encompasses three integrated components: (1) a tablet-based software that collects smoking-related information to support the development of an individualized smoking cessation quit plan and guides the ensuing text messages program; (2) a 24-week text messaging counseling program with interactive capabilities; and (3) pharmacotherapy support. Details of the interventions are described in detail in a previous publication [16].

#### 2.5.1. Decídetexto Tablet-Based Software

The tablet-based decision support tool was designed to help smokers create an individualized smoking cessation quit plan and to collect data that will tailor the text messages delivered over the ensuing 24 weeks. This tool was adapted from two smoking cessation web-based, informed decision-making tools for Latinos in the United States [17], Mexico [18], and Brazil [19]. The tablet-based tool, available in English and Spanish, consisted of interactive features that lead smokers through various steps in the process of developing a personalized quit plan. The tool included a testimony from a former smoker and features short video clips and narrated graphics on the benefits of quitting while also describing how pharmacotherapy (NRT) can support abstinence. The tool also collected participant smoking-related information, including number of cigarettes smoked per day, primary reason to quit smoking, top two smoking triggers and one strategy to manage each trigger. At the end of the 10 to 15 min tablet-based session, participants were prompted to request optional pharmacotherapy (nicotine patches or gum) and to select a quit date within a 30-day timeframe. Upon completion of the tool, participants were provided with a one-page summary print-out of their individualized smoking cessation quit plan (e.g., the selected quit date and pharmacotherapy with the recommended dose and regimen). Participants then automatically began receiving the text messaging portion of the intervention.

#### 2.5.2. Decídetexto Text Messaging

The text messaging intervention allows three levels of interactivity: (1) prescheduled standard messages, (2) keyword-triggered standard messages, and (3) counselor-personalized responses.

##### Prescheduled Standard Messages

The text messaging library consisted of 712 messages covering 10 themes: education, logistics, intra-treatment social support, coping with triggers, extra treatment social support, stimulus control, vicarious experience, relapse prevention, social norms, and reward. The text messaging system delivered these messages according to an algorithm based on fours sequential phases of the quitting process that support the personalized quit plan: (1) Pre-quit (30 days), (2) Quit-Day, (3) Post-quit Intensive (28 days), and (4) Post-quit Maintenance (20 weeks). The library also included a Relapse track (8 days).

##### Keyword-Triggered Standard Messages

These messages consisted of automated immediate responses sent to participants who text one of the following keywords: (1) Stress, (2) Crave, (3) Family, (4) Patch and (5) Gum. In addition, throughout the 24-week program, participants received 11 response-triggered (YES or NO) messages to assess their smoking status (e.g., Have you smoked a cigarette (even a puff) in the last 7 days? Text YES or NO). If participants indicated that they were smoking, these automated messages encouraged them to select a new quit date. Participants could withdraw from the text message program at any moment by texting the keyword “Stop” (e.g., “If you would ever like to stop receiving these messages, you can text STOP. However, we would prefer that you stay with us!”).

##### Personalized Responses

Taking advantage of the technological capability to recognize free texting (non-keyword) from participants, Decídetexto encouraged participants to text any concerns and/or questions to the program (e.g., “Hi, I am Francisco, your counselor. Feel free to text me anytime—I am here to help you Monday through Friday from 8 A.M. to 6 P.M. When you text me, I will reply as soon as possible”). A trained tobacco treatment specialist answered these messages following standard protocols (e.g., answering questions on pharmacotherapy delivery, use, adherence, and side effects). The tobacco treatment specialist monitored and triaged queries daily, responding within one business day of receipt of text messages.

#### 2.5.3. Nicotine Replacement Therapy (NRT)

The use of NRT in this study followed the Clinical Practice Guidelines for Treatment of Tobacco Use. NRT (nicotine patches and gum) was offered to participants at no cost. Participants who smoked more than 10 cigarettes per day (CPD) were offered 10 weeks of nicotine patches (21 mg nicotine patches to be used during the first 6 weeks, followed by 14 mg nicotine patches for 2 weeks, and 7 mg patches for the last 2 weeks). Participants who smoked between 6 and 10 CPD were offered 8 weeks of nicotine patches (14 mg nicotine patches to be used during the first 6 weeks, followed by 7 mg patches for the last 2 weeks) or gum (2 mg nicotine gum). Participants who smoked 5 or less CPD were offered 6 weeks of nicotine patches (7 mg nicotine patches) or gum (2 mg nicotine gum). Fruit chill, cinnamon, and mint flavored gum were also available. NRT was provided in two phases. At the end of the baseline assessment, participants who were interested in using NRT received a 4-week supply via postal mail for their NRT of preference. Two weeks after baseline, participants received text message prompts to request an NRT refill to continue treatment. If participants responded to the text prompts indicating interest in NRT, a 4- or 6-week supply was shipped to their home.

### 2.6. Measures

#### 2.6.1. Baseline Assessment

Trained research staff conducted an in-person baseline survey that collected sociodemographic variables such as age, gender, language of preference, education level, and annual income. Smoking-related variables collected included number of CPD, time to first cigarette, age when participants started smoking, use of menthol cigarettes, most used cigarette brand, use of other tobacco products in the past 30 days, number of quit attempts in past year, use of pharmacotherapy and/or e-cigarettes for cessation in the past, and the smoking self-efficacy questionnaire (SEQ-12). The SEQ-12 is a questionnaire measuring the confidence of smokers (current and former) in their ability to abstain from smoking in high-risk situations [20]. The SEQ-12 consists of 12 items, and each item is rated on a 5-point Likert scale (1 = not at all sure, 2 = not very sure, 3 = more or less sure, 4 = fairly sure, and 5 = absolutely sure). SEQ-12 scores range from 12 to 60 with higher scores indicating greater self-efficacy. Furthermore, we assessed body mass index, alcohol use disorder (measured by AUDIT-C: The Alcohol Use Disorder Identification Test Consumption [21]), generalized anxiety disorder (measured by GAD-2: Generalized Anxiety Disorder-2 [22]), and depressed mood (measured by PHQ-2: Patient Health Questionnaire-2 [23]).

#### 2.6.2. Month 3 Assessment

Twelve weeks after enrollment, an over-the-phone follow-up survey was conducted by trained research staff. The primary outcome was self-reported 7-day point prevalence abstinence (not smoking any cigarettes in the past seven days). Secondary outcomes were pharmacotherapy adherence, satisfaction with the intervention, therapeutic alliance, and changes in self-efficacy. Pharmacotherapy adherence was assessed by calculating the percentage of days that the participant used the nicotine patch with the number of days of prescribed medication as the denominator [24]. Consistent with Cartujano-Barrera et al., we used five categories based on the percentage of days that the participant used the patches: >75% of days, 51–75% of days, 26–50% of days, 1–25% of days, and 0% of days [25]. Satisfaction measures included questions such as “How satisfied are you with the smoking cessation program?” Therapeutic alliance and self-efficacy were measured using the Working Alliance Inventory—Short Version (WAI-S) and the SEQ-12 tests, respectively. The WAI-S is a questionnaire measuring three key aspects of the therapeutic alliance: (1) agreement on the goals of therapy, (2) agreement on the tasks of therapy, and (3) development of an affective bond [26,27]. The WAI-S consists of 12 items, and each item is rated on a 7-point Likert scale (1 = never, 2 = rarely, 3 = occasionally, 4 = sometimes, 5 = often, 6 = very often, and 7 = always). WAI-S scores range from 12 to 84, with higher scores reflecting a stronger working alliance.

#### 2.6.3. Month 6 Assessment

Twenty-four weeks after enrollment, an over-the-phone follow-up survey was conducted by trained research staff. The primary outcome was self-reported 7-day point prevalence abstinence (not smoking any cigarettes in the past seven days). Secondary outcomes were satisfaction with the intervention and social support. The Partner Interaction Questionnaire (PIQ) is the most commonly used measure of spouse/partner support related to cessation [28]. We administered the modified version of the PIQ that measures the receipt of specific behaviors from the person who follows a participant’s efforts to quit smoking most closely, not just a spouse/partner [29]. The modified version uses a 5-point Likert scale to assess how frequently the participant’s support person exhibited positive and negative behaviors. Positive items include “express pleasure at your efforts to quit”, “congratulate you for your decision to quit smoking”, and “express confidence in your ability to quit/remain quit”. Negative items include “mention being bothered by smoke”, “ask you to quit smoking”, and “criticize your smoking”. Response options are 0 = never, 1 = almost never, 2 = sometimes, 3 = fairly often, and 4 = very often. In response to the earthquake that affected Puerto Rico in January 2020, we assessed the psychosocial impact of the natural disaster. Exposure and protection after the earthquake were measured using two subscales of the Scale of Psychosocial Impact of Disasters (SPSI-D) [30]. The exposure subscale measures exposure to damage caused by the disaster, and the protection scale measures the support received after the natural disaster. The response options are 0 = highly disagree, 1 = disagree, 2 = agree, and 4 = highly agree.

#### 2.6.4. Text Messaging Interactivity

Participants’ text messaging interactions with the program were monitored throughout the entire intervention. Consistent with Cartujano-Barrera et al., participants’ text messages were categorized as keywords and free-text response to the program [31].

### 2.7. Analysis

Frequencies were calculated for categorical variables. Means and standard deviations were calculated for continuous variables. Primary analysis on smoking cessation was conducted using an intent-to-treat approach where participants lost to follow-up were considered smokers. The secondary analysis on satisfaction, therapeutic alliance, self-efficacy, and pharmacotherapy adherence were conducted using complete case analysis, in which missing values in the outcome were considered to be missing. Self-efficacy at baseline and follow-up was compared using a paired sample T-test to examine differences. A psychometric analysis was conducted to assess internal consistency and the Cronbach alpha for the WAI-S survey.

## 3. Results

### 3.1. Recruitment

A total of 38 smokers were identified. Among these, 29 were assessed for study eligibility; 26 were eligible to participate in the study. Overall, 26 smokers consented to participate and completed the baseline assessment.

### 3.2. Baseline

Baseline characteristics are displayed in Table 1. The average age of the participants was 46.8 years (SD 12.7), more than half of them (53.8%) were female, and 30.7% had a college degree. Almost all participants (96.2%) selected Spanish as their language of preference. Most participants (92.3%) smoked daily, 34.6% smoked their first cigarette within five minutes of waking up, and more than half (53.8%) used menthol cigarettes. Approximately half of the participants tried to quit smoking the previous year, and use of pharmacotherapy and electronic cigarettes for smoking cessation in the past was 34.6% and 30.8%, respectively. The mean SEQ-12 score was 39.8 (SD 12.4), which is considered to be in the moderate range of self-efficacy for quitting smoking [20]. A total of 46.1% and 15.3% of participants were classified with overweight and obesity, respectively. Four participants (15.3%) were at high-risk alcohol consumption and twelve (46.1%) screened positive for both depressive and anxiety symptoms.

### 3.3. Month 3

At Month 3, 10 participants (38.4%) self-reported 7-day point prevalence abstinence. The follow-up rate was 88.5%. All participants requested nicotine patches at baseline. Regarding pharmacotherapy adherence, 13.0% of the sample used the patch >75% of days; 26.0% used the patch 51–75% of days; 26.0% used the patch 26–50% of days; 8.7% used the patch 1–25% of days; and 26.0% used the patch 0% of days. Most participants (95.6%) reported being satisfied or extremely satisfied with the intervention. Working alliance mean value was 64.6 (SD 4.0), which is considered in the medium range of therapeutic alliance in quitting smoking [26]. The Cronbach alpha for the WAI-S survey was 0.40, which indicates a poor reliability. Self-efficacy mean scores significantly increased (*n* = 23) from 40.4 (SD 12.1) at baseline to 57.9 (SD 11.3) at follow-up (*p* < 0.01).

### 3.4. Month 6

At Month 6, 16 participants (61.5%) self-reported 7-day point prevalence abstinence. The follow-up rate was 76.9%. Most participants (90% of the sample) reported being satisfied or extremely satisfied with the intervention. Participants reported a positive ratio of perceived positive/negative support (M = 1.4). Most participants reported receiving social support from their sons and daughters (23.0%), followed by a partner (19.2%), and others (11.5%). Three participants (15.3%) reported not receiving support from anyone. Participants reported low exposure (M = 7.42, SE = 2.95) and low protection (M = 1.42, SE = 2.16) to the earthquake.

### 3.5. Text Messaging Interactivity

All participants (100%) interacted at least once with the program and sent an average of 42.9 text messages (SD 31.9; Range 1–160). Of the 1116 text messages sent by participants, only 380 (3.4%) included keywords. When analyzing the text messages sent by participants, it was noted that some participants responded to pre-scheduled standard messages as if they were interacting with a live person and not with an automatic text messaging software (e.g., “I just spent my first Christmas without cigarettes in 50 years. Thank you, God and Decídetexto”, “I already threw away all my cigarettes. I put my faith in you, you are my guides”). Moreover, participants interacted with the program to express contextual stressors (e.g., “I am sorry, I smoked. I am having a lot of anxiety due to the earthquakes”, “I am back to smoking. The earthquake has been really hard for me”).

## 4. Discussion

To our knowledge, this is the first mobile smoking cessation intervention conducted in Puerto Rico. This study demonstrates the feasibility of recruiting and enrolling Puerto Rican smokers into a smoking cessation study. Decídetexto was well received by participants, most of whom engaged in high levels of text messaging interactivity and reported high levels of satisfaction. Moreover, Decídetexto significantly increased self-efficacy, produced medium therapeutic alliance, and resulted in noteworthy cessation rates at Month 3 and 6 (38.4% and 61.5%, respectively).

It is important to acknowledge that while this study was implemented, two important contextual stressors occurred: an earthquake aid crisis and the COVID-19 pandemic. On January 7, 2020, a 6.4 earthquake impacted the southern region of Puerto Rico, rendering some areas powerless, and causing structural damages and logistic changes on the recruitment and study sites. The seismic activity lasted all of January 2020. Furthermore, due to the COVID-19 pandemic, the implementation of the quarantine implied changes in protocol for social isolation. Follow-up visits, which were initially planned to be in-person, were conducted over-the-phone. This change in the protocol, limited our ability to collect biological samples (e.g., saliva and exhaled carbon monoxide) to verify smoking abstinence. In spite of these unexpected events, participants remained highly engaged with the intervention. This finding is promising as it suggests that participants relied on the Decídetexto intervention for psychosocial support.

Participants in this pilot study reported a similar cessation rate at Month 3 to smokers in Mexico enrolled in a comparable mobile smoking cessation intervention (40%) [32]. Yet, participants in this study reported a higher cessation rate at Month 3 compared to Latino smokers in the U.S. mainland enrolled in a comparable mobile smoking cessation intervention (30.0%) [33]. A deep understanding of the differences in smoking cessation among Latino smokers living in these two different contexts (Puerto Rico and the U.S. mainland) may improve our understanding of tobacco-related disparities faced by Puerto Ricans. A smoking cessation comparative study in these two contexts can prospectively examine differences in cessation and concurrently examine potential causal pathways explaining these differences.

Puerto Rico has a local smoking cessation quitline [34,35]. However, to the best of our knowledge, the Puerto Rico quitline does not provide NRT. It is possible that the provision of NRT may have strongly influenced the outcomes of this study. The provision of NRT at no cost, as done in this study, is ongoing in several countries, including the US mainland and Canada [36,37,38,39]. This strategy, when combined with behavioral support, has demonstrated promising results in increasing cessation rates among large populations of smokers [36,37,38,39].

Similar to the study conducted in Mexico, all participants in this study requested NRT at baseline [25]. However, we found a lower pharmacotherapy adherence. Compliance with the recommended treatment adherence is significant because evidence indicates that greater medication adherence is associated with greater abstinence [40,41,42]. It is possible that the cessation rate in this study could have been higher if participants had better pharmacotherapy adherence. Research demonstrates that depression is associated with poor adherence to medical regimens, including smoking cessation treatment [43]. Moreover, people with depression might resort to dysfunctional coping mechanisms like smoking during periods of high stress [44]. As 46.1% of participants screened positive for depressive symptoms, it is imperative that future smoking cessation interventions address it. Future studies should examine reasons for suboptimal use of NRT among Puerto Rican smokers and enhance medication adherence.

Compared to the study conducted among Latino smokers in the U.S. mainland, participants in this study reported higher self-efficacy for quitting smoking at baseline [33]. However, it is important to note that in both studies, self-efficacy significantly increased from baseline to Month 3. These results highlight the importance of studying the role of self-efficacy as a predictor of smoking abstinence among Latinos. Another comparison to note is that participants in this study reported lower working alliance compared to Latino smokers in the U.S. mainland [33]. However, this result should be interpreted with caution as the questionnaire used to measure working alliance showed inadequate psychometric properties (e.g., α = 0.40). Future studies should culturally and linguistically adapt the WAI-S test.

Five studies have assessed participant interactivity in smoking cessation text messaging programs [31,32,33,45,46]. Interestingly, participants interacted at higher levels in this study (42.9 text messages per participant). Similar to the studies conducted by Cartujano-Barrera et al. and Cupertino et al., participants in this study mostly sent their own, self-composed text messages rather than relying on keywords for a program response [31,32,33]. This finding reinforces the hypothesis that text messaging interactivity via keywords may not be sufficient for smoking cessation. Future studies should assess the relationship of text messaging interactions with psychological effects (e.g., intra-treatment social support, therapeutic alliance, and perceived autonomy support).

As we implemented the study protocol, the team realized the need to culturally and linguistically adapt the intervention to Puerto Rican smokers. Some of the words used in the text messages were not popular among the Puerto Rican lexicon (e.g., *cigarillo* instead of *cigarro* (cigarettes); *parcho* instead of *parche* (patch)). Future steps include convening a Community Advisory Board in Puerto Rico to adapt Decídetexto.

This study has some limitations that should be considered when interpreting the findings. First, we only included participants who had a cellphone with a U.S. carrier (AT&T and Verizon). This eligibility criterion was implemented because the service provider of the text messaging company only had the capacity to connect with U.S. carriers. Future studies should identify a text messaging gateway that connects to all local carriers. Claro is the largest carrier in Puerto Rico [47]. Thus, participants with a Claro cellphone were not able to participate in the study. A second limitation was the small sample size. However, the sample size was sufficient to explore the feasibility and acceptability of the intervention. Third, no comparison group was available limiting our capacity to assess the efficacy of the intervention. Fourth, the sample was more highly educated than the general population in Puerto Rico. Future research is warranted to determine whether the effectiveness of the intervention is generalizable to those who are from lower socioeconomic status groups. Finally, biomarkers were not used to verify smoking status, making it possible that actual quitting rates were lower. In spite of these limitations, we found the Decídetexto intervention to be highly feasible and acceptable. The study suggests that Decídetexto holds promise for further testing, including an effectiveness analysis.

## 5. Conclusions

In Puerto Rico, the Decídetexto mobile smoking cessation intervention generated high satisfaction and frequent interactivity, significantly increased self-efficacy, and resulted in noteworthy cessation rates at Month 3 and 6, despite low adherence to NRT. Future studies should improve medication adherence among participants. Contextual stressors relevant to Puerto Rico (e.g., earthquakes aid crisis, COVID-19) should be assessed in the context of smoking cessation. Additional testing as a randomized clinical trial is warranted.

## Figures and Tables

**Table 1 ijerph-18-01379-t001:** Baseline characteristics of participants (*n* = 26).

Characteristics	*n* (%)
Age, Mean (SD)	46.8 (12.7)
Gender	
Female	14 (53.8%)
Male	12 (46.2%)
Language of Preference	
Spanish	25 (96.2%)
English	1 (3.8%)
Education Level	
Less Than High School Graduate	3 (11.5%)
High School Graduate	7 (26.9%)
Associate Degree	4 (15.4%)
Technical School	4 (15.4%)
College Graduate	7 (26.9%)
Postgraduate	1 (3.8%)
Smoking Pattern	
Non-daily	2 (7.7%)
Daily, 1–10 CPD	7 (26.9%)
Daily, 11–20 CPD	18 (69.2%)
Daily, 21 or more CPD	1 (3.8%)
Time to First Cigarette	
≤5 Minutes After Waking Up	9 (34.6%)
>5 Minutes After Waking Up	17 (65.4%)
Use of Menthol Cigarettes	14 (53.8%)
Number of Quit Attempts in Past Year	
None	13 (50.0%)
1–5 Attempts	9 (34.6%)
6–10 Attempts	2 (7.6%)
>10 Attempts	2 (7.6%)
Use of Cessation Pharmacotherapy in the Past	9 (34.6%)
Use of E-cigarettes for Cessation in the Past	8 (30.8%)
Self-efficacy for Abstinence (SEQ-12), Mean (SD)	39.8 (12.4)
Body Mass Index	
Normal Weight (18.5–24.9)	8 (30.7%)
Overweight (25.0–29.9)	12 (46.1%)
Obesity (≥30.0)	4 (15.3%)
Alcohol Use Disorder	
Deny Any Alcohol Consumption: AUDIT-C 0	11 (42.3%)
Lower Risk Drinking: AUDIT-C 1–3 (men), 1–2 (women)	4 (15.3%)
At-risk Drinking: AUDIT-C 4–7 (men), 3–7 (women)	7 (26.9%)
High Risk Drinking: AUDIT-C ≥ 8	4 (15.3%)
Depression	
Positive (PHQ-2 ≥ 3)	12 (46.1%)
Negative (PHQ-2 < 3)	14 (53.8%)
Anxiety	
Positive (GAD-2 ≥ 3)	12 (46.1%)
Negative (GAD-2 < 3)	14 (53.8%)

AUDIT-C: Alcohol Use Disorders Identification Test—Concise; CPD: cigarettes per day; GAD-2: Generalized Anxiety Disorder—2; PHQ-2: Patient Health Questionnaire—2. SEQ-12: Smoking Self-Efficacy Questionnaire.

## Data Availability

The datasets generated for this study are available on request to the corresponding author.

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
