# Peer review of "Decídetexto: Feasibility and Acceptability of a Mobile Smoking Cessation Intervention in Puerto Rico"

_ijerph, 2021, doi:10.3390/ijerph18041379_

Round 1
Reviewer 1 Report
This is an interesting and well-written manuscript presenting findings from the pilot study with 26 smokers that were subject to a mobile smoking cessation intervention in Puerto Rico. The authors aimed at assessing the feasibility and acceptability of a mobile smoking cessation intervention among Puerto Rican smokers. Smoking remains an important health issue in Puerto Rico, and is even more prominent among Puerto Ricans living in mainland U.S. Socio-economic status (SES) of the latter group makes it hard to reach with smoking cessation interventions, and literature provides little suggestions for effective public health interventions in this regard.
The biggest advantage of this paper is the fact, that the findings from this study may provide a good starting point for larger trial intervention, preferably with a control group.
However, to improve the quality of the paper (and potential implementation of the presented findings), some remarks are needed:
General remarks:
- Authors extensively present the differences between the smokers population in Puerto Rico and smoking Puerto Ricans in the mainland U.S., as well as between Puerto Ricans and other ethnic groups (lines 37-45). This leads to the statement that there is a need for studies to understand and address the unique tobacco-related disparities among Puerto Rican smokers in the U.S. mainland and Puerto Rico (lines 48-50) - which is beyond dispute. However, the presented study does not aim at that issue (lines 57-59), and it is not discussed further in the text.
To maintain consistency some linkage between those two statements should be provided – showing that there is a need for studies on effective smoking cessation interventions in Puerto Rico which could be implemented also among Puerto Ricans in the U.S. mainland.
- As Decídetexto incorporates, as one of its components, pharmacotherapeutic support – free of charge delivery of NRT (line 154), information on rules concerning access to such therapy in Puerto Rico would be very helpful (e.g. is NRT free for the general public?). If NRT is not widely accessible (due to financial or administrative reasons), free provision of nicotine patches/gums might strongly influence the outcome of the trial.
This information can be provided in section 1. Introduction or section 4. Discussion (lines 319-330).
- A separate section (conclusion) summarizing the findings and implications of the study would be advisable.
Detailed comments:
- Materials and Methods line 70 - please provide Ethical Board consent number (alternatively this could be done in lines 380-382),
- Materials and Methods line 111 – term “NRT” has been already explained in lines 63-64,
- Materials and Methods line 195-196 – please provide information on what scale was used to assess general satisfaction of participants (as in the case of PIQ),
- Materials and Methods line 218-219 – please consider rewriting this sentence as it is not clear,
- Results line 252-253 - please provide a reference for the statement on SEQ-12 score considered as “moderate range of self-efficacy for quitting smoking”,
- Results line 264-266 - please provide a reference for the statement on working alliance value in quitting smoking,
- Results line 281 – decimal point placement is incorrect,
- Discussion - please provide brief information about smoking prevalence in Puerto Rico and some demographic/SES insights on smokers population,
- Discussion lines 255 and further – as participants of the study did not reflect (in terms of SES) the average Puerto Rican smokers, this limitation shall also be listed in the conclusion section.
Author Response
Dear reviewer,
We are pleased with the noted strengths of our manuscript entitled “Decídetexto: Feasibility and acceptability of a mobile smoking cessation intervention in Puerto Rico”. Below we respond to reviewers’ recommendations. Changes in the manuscripts are highlighted in yellow.
- As Decídetexto incorporates, as one of its components, pharmacotherapeutic support (NRT) at no cost, information on rules concerning access to such therapy in Puerto Rico would be very helpful (e.g., is NRT free for the general public?). If NRT is not widely accessible, free provision of nicotine patches/gums might strongly influence the outcome of the trial.
This is a very good point. We have included the following paragraph in the Discussion section: “Puerto Rico has a local smoking cessation quitline [34-35]. However, to the best of our knowledge, the Puerto Rico quitline does not provide NRT. It is possible that the provision of NRT may have strongly influenced the outcomes of this study. The provision of NRT at no cost, as done in this study, is ongoing in several countries, including the US mainland and Canada [36-39]. This strategy, when combined with behavioral support, has demonstrated promising results in increasing cessation rates among large populations of smokers [36-39].”
- A separate section (conclusion) summarizing the findings and implications of the study would be advisable.
We have included a Conclusion section. “In Puerto Rico, the Decídetexto mobile smoking cessation intervention generated high satisfaction and frequent interactivity, significantly increased self-efficacy, and resulted in noteworthy cessation rates at Month 3 and 6, despite low adherence to NRT. Future studies should improve medication adherence among participants. Contextual stressors relevant to Puerto Rico (e.g., earthquakes aid crisis, COVID-19) should be assessed in the context of smoking cessation. Additional testing as a randomized clinical trial is warranted.”
- Please provide Ethical Board consent number.
We have included the protocol number (1906014547).
- Line 111 – term “NRT” has been already explained in lines 63-64.
Thank you! We have removed the explanation of the NRT in line 111.
- Line 218-219 – please consider rewriting this sentence as it is not clear,
We have rewritten the sentence. “In response to the earthquake that affected Puerto Rico in January 2020, we assessed the psychosocial impact of the natural disaster.”
- Line 252-253 – please provide a reference for the statement on SEQ-12 score considered as “moderate range of self-efficacy for quitting smoking”.
We have included the reference.
- Line 264-266 – please provide a reference for the statement on working alliance value in quitting smoking,
We have included the reference.
- As participants of the study did not reflect (in terms of SES) the average Puerto Rican smokers, this limitation shall also be listed in the conclusion section.
We have specified in the Limitations that “…the sample was more highly educated than the general population in Puerto Rico. Future research is warranted to determine whether the effectiveness of the intervention is generalizable to those who are from lower socioeconomic status groups.”
Kindest regards,
Francisco Cartujano-Barrera, MD
Research Assistant Professor
Department of Public Health Sciences
University of Rochester Medical Center

Reviewer 2 Report
Dear Authors,
The manuscript named “Decídetexto: Feasibility and acceptability of a mobile smoking cessation intervention in Puerto Rico” does an excellent job addressing a critical problem in health disparities. Thank you for your hard work. I have reviewed your manuscript and added my comments to all the relevant sections below.
Major criticisms
- The authors do not state the research hypothesis in the manuscript. Please, include the hypothesis in the manuscript and assert if the hypothesis was verified or falsified at the end.
- The abstract does not state the purpose of the study. Therefore, people reading your work will not gather the purpose at first glance. Please, include the purpose of the study in this section.
- Need to address and state the possibility of selection bias in the limitations paragraph.
Minor criticisms
Tables
Table 1. lacks footnotes and abbreviations clarifications.
English issues:
Replace: "U.S. mainland" for "the U.S. mainland" at lines 44, 312, 314, 331, and 337.
Replace: “breast-feeding” for “breastfeeding” line 87.
Replace: “the importance to study” for “the importance of studying” line 334.
Replace “reported higher cessation rate” for “reported a higher cessation rate” line 311.
Replace “criteria” for “criterion” in line 357.
Author Response
Dear reviewer,
We are pleased with the noted strengths of our manuscript entitled “Decídetexto: Feasibility and acceptability of a mobile smoking cessation intervention in Puerto Rico”. Below we respond to reviewers’ recommendations. Changes in the manuscripts are highlighted in yellow.
- The authors do not state the research hypothesis in the manuscript. Please, include the hypothesis in the manuscript and assert if the hypothesis was verified or falsified at the end.
We have not included a hypothesis as, in accordance with Leon, Davis, and Kraemer, “A pilot study is not a hypothesis testing study” [1].
- The abstract does not state the purpose of the study. Therefore, people reading your work will not gather the purpose at first glance. Please, include the purpose of the study in this section.
We have included the purpose of the study in the abstract.
- Table 1. lacks footnotes and abbreviations clarifications.
We have included the footnotes and abbreviations clarifications. Thank you!
- Replace: "U.S. mainland" for "the U.S. mainland" at lines 44, 312, 314, 331, and 337.
We have made the replacement.
- Replace: “breast-feeding” for “breastfeeding” line 87.
We have made the replacement.
- Replace: “the importance to study” for “the importance of studying” line 334.
We have made the replacement.
- Replace “reported higher cessation rate” for “reported a higher cessation rate” line 311.
We have made the replacement.
- Replace “criteria” for “criterion” in line 357.
We have made the replacement.
Reference:
- Leon AC, Davis LL, Kraemer HC. The role and interpretation of pilot studies in clinical research. J Psychiatr Res. 2011;45(5):626-629.
Kindest regards,
Francisco Cartujano-Barrera, MD
Research Assistant Professor
Department of Public Health Sciences
University of Rochester Medical Center
